# Sex Differences in Biological Systems and the Conundrum of Menopause: Potential Commonalities in Post-Menopausal Disease Mechanisms

**DOI:** 10.3390/ijms23084119

**Published:** 2022-04-08

**Authors:** David A. Hart

**Affiliations:** 1Department of Surgery and Faculty of Kinesiology, University of Calgary, Calgary, AB T2N 4N1, Canada; hartd@ucalgary.ca; Tel.: +1-403-220-4571; 2Bone & Joint Health Strategic Clinical Network, Alberta Health Services, Edmonton, AB T5J 3E4, Canada

**Keywords:** menopause, evolutionary consequences, common mechanisms, medical implications, endothelial cells, microvasculature regulation

## Abstract

Sex-specific differences in biology and physiology likely start at the time of conception and progress and mature during the pre-puberty time frame and then during the transitions accompanying puberty. These sex differences are impacted by both genetics and epigenetic alterations during the maturation process, likely for the purpose of preparing for successful reproduction. For females, later in life (~45–50) they undergo another transition leading to a loss of ovarian hormone production at menopause. The reasons for menopause are not clear, but for a subset of females, menopause is accompanied by an increased risk of a number of diseases or conditions that impact a variety of tissues. Most research has mainly focused on the target cells in each of the affected tissues rather than pursue the alternative option that there may be commonalities in the development of these post-menopausal conditions in addition to influences on specific target cells. This review will address some of the potential commonalities presented by an integration of the literature regarding tissue-specific aspects of these post-menopausal conditions and data presented by space flight/microgravity (a condition not anticipated by evolution) that could implicate a loss of a regulatory function of the microvasculature in the risk attached to the affected tissues. Thus, the loss of the integration of the paracrine relationships between endothelial cells of the microvasculature of the tissues affected in the post-menopausal environment could contribute to the risk for post-menopausal diseases/conditions. The validation of this concept could lead to new approaches for interventions to treat post-menopausal conditions, as well as provide new understanding regarding sex-specific biological regulation.

## 1. Introduction

The development, maturation and senescence of humans requires them to undergo a series of gradual and event-related biological “steps” across the lifespan. Thus, many biological systems arise during development that distinguish males and females. Obviously, there are those associated with the differences in functions of the two sexes, but there are also other “secondary” systems that also differ. While many more biological systems exhibit differences after puberty, and there are active epigenetic alterations in some systems as a consequence of the transition through puberty [1,2,3,4]. However, differences in some systems, such as the immune system and inflammatory responses, appear to be evident before and after puberty. The female preponderance of many autoimmune diseases, such as rheumatoid arthritis (an inflammatory autoimmune disease), in females compared to men may be due to sex differences in immune regulation [5,6,7].

However, it should be pointed out that only a small percentage of females develop autoimmune diseases, and, thus, the advantages associated with any sex differences must outweigh the risk of adverse situations. However, this example does serve to indicate that females are heterogeneous with respect to sex differences. Another example was reported from studies investigating changes in knee joint laxity during the menstrual cycle where 20% of the young females in the study did not exhibit changes in joint laxity even with similar changes in estrogen levels [8,9,10]. In addition, some females retain fluids during parts of the menstrual cycle, but other females do not [11,12]. Furthermore, as will be discussed below, many females will develop the risk of some conditions after menopause, but many will not and those at risk are not at risk for all of the conditions. Therefore, some sex differences can be generalized, but on an individual level, the population is very heterogeneous.

## 2. Biological Sex Differences and Function

For many of the biological systems that appear to exhibit sex differences, such as the immune and inflammatory systems, the differences are believed have arisen during evolution and are associated with successful pregnancies and the survival of offspring. The conception and implantation of the fetus, what is effectively an allogeneic “graft”, is enhanced by histoincompatibility [13,14,15], but it also requires the mother to regulate the process so as not to eliminate the fetus. Interestingly, many females (~70%; [16]) with the autoimmune inflammatory disease rheumatoid arthritis (RA) undergo remission of their disease while pregnant [17,18,19,20], indicating that there can be a down regulation of aspects of inflammatory responses during established pregnancy. Why this does not apparently occur for all females with RA is not known, but it again speaks to heterogeneity. Furthermore, menopause can also influence inflammatory autoimmune diseases such as RA [21], which are predominately diseases of females, and the treatment of such diseases with drugs such as glucocorticoids can also impact post-menopausal conditions such as osteoporosis [22].

A number of other sex-associated differences in biological systems also appear to be related to successful pregnancy or the post-natal survival of the offspring during lactation. These include calcium mobilization from bone to meet the needs of the developing fetus, transient obesity or enhanced energy storage to meet the needs of both the mother and the fetus, particularly in times of food insecurity, and alteration in cardiovascular regulation to meet the demands exerted by the development and maturation of the fetus. During the lactation period, immune cells/lymphocytes capable of making antibodies to intestinal pathogens migrate from gut mucosal immune depots to breast tissue and, as a consequence, protective antibodies appear in the milk to enhance the survival of the baby from intestinal pathogens [23,24]. In early evolutionary times, when hygiene may have been less stringent than it is currently in many western societies, this mechanism could have offered a significant survival advantage. Similarly, continued access to calcium stores for the milk would also enhance growth during this critical time for the offspring. Of importance, many of these pregnancy and lactation-associated changes are reversible, and most females appear to be able to re-adapt back to the pre-pregnancy state [25]. Thus, during the reproductive stage of the lifespan, sex-associated differences in biological regulation serve important “survival of the species” functions [13,18,26]. It is of interest that if these sex-associated differences were so critical, why is there such heterogeneity in female populations for the regulation of these functions (i.e., why do only a subset of females develop any one of the potential post-menopausal conditions/diseases)? The answer to this question is not clear and could have several explanations, but the question is also reinforced by the heterogeneity in the response to menopause of females discussed below.

## 3. Menopause

Menopause is the transition that every female goes through at ~45–55 years of age. Thus, after ~35+ years of menstrual cycles, females undergo a process that occurs over several years and culminates in the cessation of menstrual cycles and loss of ovarian function [27,28,29,30]. As a consequence of menopause, females become more at risk for a number of conditions/diseases such as osteoporosis, cardiovascular disease, dementia, obesity, sarcopenia and osteoarthritis [31], and several of these are discussed in detail below.

While nearly all females undergo natural menopause (some undergo surgical menopause due to disease), an individual woman does not exhibit risk for all of the above-mentioned conditions/diseases, although they may develop more than one of the risks. Furthermore, even within the population that develops a condition such as osteoporosis, there is considerable variation in the rate of bone loss that can be several-fold in extent [32,33,34]. Thus, prior to menopause, the medical implications of these conditions are apparently obscured by hormones such as estrogens and progesterone. Therefore, this temporal relationship has led to people to assume that the systems involved are related to reproduction either directly or indirectly, as discussed above.

## 4. Why Menopause?

The question of why menopause would have been selected for during evolution has been addressed and reviewed by a number of researchers [27,28,29,30]. Some of the hypotheses relate to sociological advantages of older and experienced females who would then have had more time to help with younger children and mothers. However, it is not clear how such a sociological use would be translated to biological loss. Alternatively, it could relate to the disadvantage of older females having children who did not contribute to the survival of family/clan groups due to trisomy and other abnormalities of fetal development [35]. As most females of ~45 years of age do not actually have many viable “eggs” left compared to when they were born [36], the combination of egg numbers, increased risk for children with chromosomal abnormalities and a lifespan that likely was <45 for much of the critical evolutionary history, a biological explanation might be more relevant than sociological considerations, particularly when biological issues likely predate sociological developments.

## 5. The Consequences of Menopause

When the average lifespan began to exceed 45 years somewhere in documented history, it became apparent that females underwent a cessation of ovulation and menstrual cycles. This was not likely observed through early evolutionary history when the lifespan was much shorter due in part to food insecurity, deaths in childbirth, infectious diseases, accidents and predation. Interestingly, in modern times, there are some features of menopause that are quite common (~80%), such as hot flashes/vasomotor disturbances [37,38,39] and “brain fog”/cognition changes [40,41]. Whether these early symptoms reflect the risk of later conditions is not well documented, but smaller subsets of post-menopausal females subsequently experience a variety of conditions that can pose serious health threats. While the risk of post-menopausal conditions may have some genetic basis [42], how such risk is manifested is yet to be determined. However, several of these post-menopausal conditions are discussed below.

### 5.1. Osteoporosis

As discussed above, there are several conditions/diseases that can appear in a subset of females in the post-menopause part of the lifespan. While not all females develop all of the indicated diseases, even in the context of the individual conditions/diseases, considerable variation in disease occurs, as assessed by the rate of dysfunction after menopause or the extent of the condition. Thus, there is the risk of developing one or more of the conditions/diseases following menopause, but the risk is not absolute and must, therefore, be dependent on other factors for the actual elaboration of the risk. Using osteoporosis as an example, the disease “runs in families” and therefore likely has a genetic underpinning [43], but the rate of disease progression and the extent of bone loss can vary extensively and can influence diagnosis [44,45,46]. Therefore, in the pre-menopausal state, such variation does not negatively affect the use of that system, otherwise the variation would likely have been lost during evolution. Thus, there is likely a positive effect associated with the risk in the pre-menopausal state, or the mechanistic basis for the variation may be silent during the reproductive years. Furthermore, as only a subset of post-menopausal females develop clinical osteoporosis, many women are not affected by that state, and therefore the molecular basis for the osteoporosis risk itself is not required for the function in the pre-menopausal state.

Interestingly, there is evidence that bone metabolism can be influenced during the menstrual cycle [47,48,49,50], pregnancy [51,52] and lactation [25,52,53]. A very interesting aspect of the bone loss associated with the obtainment of Ca^2+^ for fetal development and post-natal growth is that it can apparently be replaced over time. Thus, for bone and calcium availability, there is perhaps a teleological reason behind the risk for osteoporosis in the post-menopausal state. However, it also implies that there is a hormonal influence regarding the recovery phase post-pregnancy and post-lactation [51,52,53]. Interestingly, the variation in rate or extent of bone loss after menopause, the risk of which is “silent” prior to menopause, does not seem to compromise or enhance the functioning of the system. The caveat to this conclusion is that there have not been any detailed reports of individual variations in bone turnover during pregnancy and lactation and recovery to determine whether there are correlations with the rate of progression or extent of bone loss after menopause, although some correlations have been noted in bone quality [54] or associations between lactation and subsequent osteoporosis [55]. Finally, the possibility that the variability is only evident when the hormonal changes are no longer present, and the variation is only due to the post-menopause environment should also be entertained.

Finally, it should not be forgotten that bone is innervated [56,57,58], and, therefore, could also be regulated by neural input that changes after menopause in addition to the direct effects of a hormone such as estrogen on bone cells. In fact, all of the systems “adversely” influenced by menopause are innervated to varying degrees. Potentially, cartilage is the exception to that rule, but osteoarthritis development post-menopause may initially be due to alterations in tissues of the knee other than cartilage (considering that the joint is an integrated organ and defects in one component can lead to the loss of cartilage secondarily [59,60]). However, it is not clear how innervation alterations would lead to variable loss of bone after menopause. In fact, it is not entirely known how such variable loss of bone is manifested!

Finally, it should be noted that osteoporosis (OP) is not just a disease affecting post-menopausal females, with ~70–75% of cases being females, but also occurs to varying extent in aging males [61,62]. Furthermore, bone loss can also occur as a consequence of being in space and removed from the 1 g environment of Earth [63,64,65], an environment that could not have been anticipated during the evolution of humans. Interestingly, most astronauts are young to middle-aged males, and the bone loss in space is greater in the bones of the lower extremities than those of the upper extremities. Of relevance to the present discussion is the finding that bone loss in astronauts is also quite individually variable, with losses of 0.1–2% per month, and not unlike the variable loss in the subset of post-menopausal females developing osteoporosis discussed previously. Such correlations may indicate that actual bone loss variation is due to properties/characteristics of the bone cell system and perhaps not of the initiating mechanism associated with menopause. Prolonged bedrest on Earth can also lead to bone loss, with similar variable rates to those observed in space [66,67], and the initiation of accelerated bone turnover can occur rapidly [68]. Therefore, even on Earth, bone homeostasis is regulated by the “use it or lose it” principle [65]. However, reports on how bone loss is regulated in post-menopausal females with OP who are then subjected to prolonged bedrest could not be found in the literature.

Interestingly, the findings to date also indicate that for post-menopausal females developing OP, the mechanisms involved appear to override the influence of the 1 g environment on bone regulation, as the OP population is still ambulatory and subjected to ground reaction forces while they walk, run or exercise. Thus, either the risk for OP develops at the same time as the implementation of the 1 g regulatory mechanisms or, if the risk for OP is implemented later (i.e., at puberty), such influences can override a component of the mechanisms that regulate the 1 g response elements.

### 5.2. Dementia

With the aging of population, particularly in developed countries, there has been an increasingly large number of individuals displaying various forms of dementia (~1–8% of the population over age 60). Dementia and the loss of cognition currently affect ~5.2 million Americans [69] and many more in other developed countries. The numbers in countries such as the United States of America are projected to double by 2050 [69]. Interestingly, ~2/3 of those affected are post-menopausal females [70], a percentage not too dissimilar to the percentage of those affected by osteoporosis. Furthermore, it has been reported that women who have had bilateral oophorectomy are at increased risk of dementia [71], again linking the risk for development of dementia to the loss of ovarian hormones.

While the causes of early and late dementia may be varied, some forms appear to be more vascular related than others [72,73,74]. Some forms may also have a genetic or epigenetic risk as well [74]. A concerted research effort by the pharmaceutical industry and many academics has been focused on developing effective interventions to prevent or slow progression, and this has been a challenging endeavor. Interestingly, some research has indicated that exercise in midlife and in the post-menopausal state can exert beneficial effects on the conditions [75,76,77,78,79], particularly those of a vascular nature [76]. As the pathology associated with some of the forms of dementia may be related to the tau protein and other brain proteins (discussed in [80,81,82,83]) and involve specific regions of the brain (i.e., the hippocampus and hypothalamus [81,82,83,84]), the presence of this pathology may be a consequence of the disease initiation but not the initiating cause. However, the responsiveness to exercise may imply that there is a loss of vascular-mediated integrity with aging in these individuals, a loss that can either be prevented by exercise or impacted by exercise. The form of exercise that is effective may be individual in nature, as it is known that not all individuals respond to aerobic and resistance exercise in the same way [64]. In addition, it is not well defined whether the effect of exercise is a direct effect of the exercise on the vascular cells, perhaps via blood pressure or shear-related gene effects, or an indirect effect via the release of mediators from muscles activated by the exercise [64]. Candidate mediators include irisin, which is known to be released from activated muscles and can influence brain centers in animal studies [85,86,87] and has been implicated in humans as well [88]. A second potential exercise-dependent mediator is clusterin [89]. However, it is not known whether these affect brain centers by directly impacting end target cells in these tissues or via interactions with the endothelial cells of the affected centers or both.

### 5.3. Cardiovascular Disease

Prior to menopause, females appear to be more protected from the development of cardiovascular disease compared to age-matched male populations [90,91,92,93]. However, after menopause, this protective effect of hormones is apparently lost for a subset of females, who are at increased risk of cardiovascular events [93,94,95]. Again, not all post-menopausal females appear to exhibit this risk and, thus, again there is heterogeneity in the female population regarding this risk. The basis for this heterogeneity is not well defined, but this aspect of post-menopausal life has been the subject of a number of recent reviews in addition to those referenced above [96,97,98].

As some aspects of the risk of cardiovascular disease may also develop with aging, such as increasing vascular stiffness [99], the increased risk for a subset of females after menopause is also in conjunction with potential aging processes. Interestingly, some aspects of increases in arterial stiffness in post-menopausal females can be impacted by exercise programs [94]. Thus, exercise can prevent or alleviate some of these post-menopausal features, implicating an influence on the vascular system. Furthermore, exercise can also influence the loss of cognition in older individuals in a sex-specific manner [76,77,100,101], indicating perhaps some commonalities between sex hormones, vascular integrity and cognition loss [102]. Interestingly, one of the features of certain forms of dementia is the accumulation of tau in areas of the brain, and the enhanced expression of tau can also occur in endothelial cells of the brain [103]. Thus, in a subset of older post-menopausal females, there may be common links between a dysregulated microvasculature and some forms of dementia at a mechanistic level.

Finally, alterations in vascular integrity have also been noted in astronauts exposed to microgravity [104,105,106], including sex differences [107]. In addition, using the prolonged bedrest surrogate for spaceflight, changes have also been documented in females [108,109]. As most astronauts have been males, and it has been postulated that exposure to microgravity is a mimic of aging [110], the situation in post-menopausal females at risk for such changes may be a unique subset of females. Interestingly, and relevant to the previous discussion regarding variation in the extent of bone loss in space microgravity, there is also variation in changes in cardiovascular conditioning in astronauts, and, thus, there may be more than one mechanism leading to such changes in astronauts versus post-menopausal females. However, it is not yet known whether the cardiovascular alterations occurring during bedrest for young women predict any post-menopausal correlations, and this area may be fruitful to explore.

### 5.4. Obesity

Many females gain significant amounts of weight during pregnancy, while others do not. While some of this may relate to what and how much is eaten, for others it may also relate to activity levels and other factors. After pregnancy, many females lose the extra weight without effort, but this is variable. In the post-menopausal state, a subset of females appears to gain weight in specific sites on the body [111], and this age-related weight is located differently in males and females [112,113], but there are reported racial differences in the deposition of fat [114,115]. Certainly, with the advent of the current “obesity epidemic” (discussed in [116,117]), a large percentage of females and males in many developed countries are overweight or suffer from obesity. Whether the mechanisms for the latter (i.e., diet and sedentary behaviour) are the same or different for the weight gain following menopause is not clear, but it is likely that it is different since it is related to hormone loss and was apparent in females long before the current obesity epidemic.

One could potentially consider this risk for post-menopausal obesity as an advantage presented by an enhanced ability to store energy during pregnancy. However, pregnancy is a fairly acute event of ~9 months duration, while in the post-menopausal state, the weight gain can become chronic [117], possibly leading to changes in both adiposity and the ability to shed fat by exercise and dietary considerations. Unfortunately, in the modern era, some of the individual factors may overlap, further clouding the interpretation and elucidating of the mechanisms involved.

### 5.5. Osteoarthritis

Osteoarthritis (OA) is an umbrella term used to describe a degenerative process in a joint such as the knee or hip [118]. Thus, OA may encompass a number of disease subsets such as post-traumatic OA resulting from an overt joint injury or metabolic OA associated with obesity [118]. However, it is also considered a disease of mechanics [119], where the biomechanical integrity of a multi-tissue joint organ system [59,60] becomes compromised. Prior to the age of menopause, the incidence of OA of the knee or hip is very similar in males and females. However, after the age of menopause, sex-associated differences in incidence develop, with the female to male ratio exceeding 2/1 [118]. Thus, after menopause, a subset of females exhibit a high risk of developing what is termed “idiopathic” OA or OA that cannot be attributed to other causes. This risk does not appear to be related to obesity or other known factors, although obesity is certainly a risk factor of its own [120,121,122]. The questions of why such a risk of OA should have been maintained in the population, why only a subset of females exhibit such a risk and what fundamental mechanisms are involved remain to be elucidated. Certainly, the connective tissues of joints such as the knee express receptors for estrogen and progesterone [123], and joint functioning can be affected by both the menstrual cycle [8,9,10] and pregnancy [124,125,126], with some changes persisting after pregnancy [126]. Therefore, these tissues are hormone responsive, but why only a subset of post-menopausal females are at risk of OA must be due to factors in addition to the loss of ovarian hormones, perhaps due to factors related to growth and development variables [118], or potentially due to a subset that may have pregnancy-associated changes that persist and/or epigenetic changes in the cartilage or one of the other joint tissues. While articular cartilage, a tissue often central to OA development, expresses estrogen receptors [127,128,129], it is not vascularized or innervated, and thus it is in a unique position to be either affected directly in the post-menopausal state or indirectly due to alterations in the regulation of the other joint tissues.

As mentioned in the above discussion, subsets of post-menopausal females are at risk for a spectrum of conditions or diseases. Not all females experience the same risk of an individual condition, and not all females experience risk in terms of all of the conditions. While much of the risk is currently not well defined, it is the subject of much research and there are some patterns which could be addressed. Some of these are discussed below and are focused on addressing the question “Are the diseases or conditions that arise in the post-menopausal state all a result of completely different mechanisms, or do they arise from similar or related mechanisms in different tissues?”

While the diseases/conditions of menopause are often considered by many to be independent conditions, it is also clear that in a certain population, one condition can increase the risk for others. That is, obesity and its consequences, such as metabolic syndrome, can also increase risk for cardiovascular disease [130,131], osteoarthritis [132] and dementia [133,134]. Furthermore, immune cells can also interact with the skeletal system [135] and this may have further relevance in the post-menopausal state [136]. Thus, in the post-menopausal state, some interactions between factors could have secondary consequences involving several systems.

Interestingly, the way that medicine is taught and practiced aligns medical doctors with body parts and disease entities. Thus, while family physicians are well positioned to treat the whole patient, they often function as gatekeepers [137,138], and patients with specific post-menopausal conditions are treated by specialists. For example, those with osteoporosis are treated by endocrinologists, whereas those with heart and cardiovascular system disorders are treated by cardiologists, osteoarthritis is in the purview of orthopedic surgeons and rheumatologists, obesity in that of internal medicine or family practice experts, menopause in the purview of endocrinologists and gerontologists and dementia that of neurologists. Furthermore, individuals are also segregated by life stage, with children and adolescents the focus of pediatricians and the elderly by a variety of specialists depending on any comorbidities, and there can be challenges associated with such age transitions [139,140]. Therefore, there is an associated limitation with this paradigm to focus on the target tissue that is directly associated with the symptoms, rather than look for any commonalities. In perspective, this is not entirely negative, in that one does not have to know the mechanism(s) which cause the disease to develop effective treatment interventions. However, the approach may overlook some salient features and questions, and thus integrating the various pieces of this post-menopausal puzzle may yield a new understanding of the disease constellation in the post-menopause state. Until proven otherwise, it is certainly convenient to address the conditions of menopause as separate conditions or diseases that arise following the cessation of ovarian function, menstrual cycles, pregnancy risk and the loss of the regulatory activities of hormones such as estrogen and progesterone. However, if that is the case, a number of issues and questions arise, and several of these are addressed below.

(1)One may assume that in the pre-menopausal state, the potential risk for the various post-menopausal conditions must serve a purpose in that they arose and have been retained in the genome or epigenome. If retained for a purpose, then most likely the purpose is associated with reproduction advantages. Thus, all of the post-menopausal conditions could be related to various aspects of reproduction. If that is the case, then why do only a subset of females develop OP, dementia, OA or cardiovascular disease, and, thus, would those with such post-menopausal risks not be selected for over millennia and a higher percentage of females have most or all of the conditions?(2)Is the fact that most females do not express all of the post-menopausal conditions indicated above due to adverse outcomes if the risks of all are evident in the pre-menopausal states of individuals? In fact, as a large percentage of females in developed countries now have chronic obesity in the decades prior to menopause, has this altered risk for other post-menopausal conditions compared to during evolutionary history? Therefore, could some of the post-menopausal conditions be conditions of more recent life, with them not being a “risk” when these conditions arose?(3)Thanks to the extensive evidence regarding bone loss under various conditions, the post-menopausal condition of OP may be used as a focal point for discussion of alternative explanations for post-menopausal conditions. Some of the most relevant pieces of evidence are outlined below in point form:(A)Astronauts in microgravity lose bone at a variable rate and from the lower extremities > the upper [141,142]. As most astronauts have been males, this means that when removed from a 1 g environment, males can lose bone, but the rate is quite variable, similar to post-menopausal females with OP (~75% of OP patients on Earth).(B)Young skeletally mature males and females subjected to prolonged bedrest under controlled conditions lose bone at a variable rate [67]. Thus, the failure to experience bone ground reaction forces (GRFs) in the presence of a 1 g environment leads to the loss of bone and this is an example of the “use it or lose it” principle [65,143,144]. As both males and females lose bone at a variable rate, both males and females exhibit variation in bone loss when a 1 g environment is lost or GRFs are not engaged, but only a subset of females develop OP after menopause. Thus, the bone cell targets in OP can be dysregulated by both the loss of mechanical engagement and after menopause. Does that imply separate regulatory systems?(C)Related to (B), only a subset of females develop osteoporosis after going through menopause or after being subjected to ovariectomy and the loss of ovarian sex hormones. The rate of bone loss is variable in different individuals and may have secondary consequences [145], and not all sites/locations lose bone equally. From bedrest outcomes, it can be assumed that females who do not develop OP after menopause likely also have the potential for variable bone loss if removed from GRFs and, potentially, when exposed to microgravity. Thus, the subset of females developing OP must have a defect in a regulatory system that can override the mechanisms behind the mechanical stimulation (i.e., GRFs in a 1 g environment), and variable bone loss is somehow intrinsic to the end stage target cells (i.e., bone cells) but not a putative regulatory system. Furthermore, as the loss of mechanical stimulation and/or a microgravity environment appears to lead to more bone loss in lower extremities and this is not the case in OP, the putative regulatory system regulated by sex hormones such as estrogen and estrogen receptors is likely not dependent on a mechanical stimulus. Furthermore, it may also not be dependent on a mechanical stimulus for those males who develop OP (~25% of the cases), as they also are physically active and subjected to ground reaction forces even when they have the disease.(D)The loss of ovarian sex hormones associated with menopause likely does not lead to reversion to a state that existed prior to puberty. This conclusion is based on the fact that after puberty, skeletal maturity is established and thus a biological set point is entrenched. Secondly, puberty is associated with extensive epigenetic modifications [2,3,4], and thus such modifications and any others occurring due to life experiences are not reversed by menopause and, after ~35 years of menstrual cycles, only interrupted by pregnancies. The pre-puberty years are associated with growth and maturation, while puberty is associated with biological modifications associated with sex hormone elevations and in females the preparation for reproductive success. This scenario raises the possibility that the risk of OP and potentially other post-menopausal conditions/diseases arise as a consequence of puberty, possibly linked to variations in the targeting of epigenetic modifications associated with puberty and sex hormone receptor activities.(4)If one takes the contrarian view that many or most of the post-menopausal conditions have a common underlying mechanistic feature related to hormones and hormone action, what could this be? First, all of the affected tissues or systems are vascularized and innervated except for articular cartilage. While cartilage is neither and is central to OA, the articulating joints function as organ systems. Furthermore, there are different sub-types of OA and cartilage may just be the “weak link” due to the fact that it is devoid of vascularity and innervation [118].

The above list of characteristics and issues identified regarding OP development after menopause leads to the conclusion that the condition likely does not depend solely on influences of menopause on the target bone cells. Therefore, using OP as an example and supportive evidence from other conditions arising post-menopause, commonalities in post-menopausal conditions could relate to the regulatory functions of endothelial cells in the affected tissues and/or the innervation. In many tissues, the innervation parallels the microvasculature and can contribute to the regulation of the cells of the microvasculature, particularly during pregnancy [146,147], a point which is relevant to this discussion as this relationship implies some interdependency regarding regulation in the vasculature.

Secondly, the microvasculature in an organ or system is likely differentiated to perform unique roles in the context of that organ or system in a paracrine manner while retaining characteristics required to function in the microvasculature [148,149,150]. Such heterogeneity has been noted for the bone [151], the materno–fetal endothelial barrier [152] and the brain [153,154,155] and has been postulated to be mechanistically involved in brain pathology [156], and the vasculature may be a target for neurological disorders [157]. Interestingly, in a preclinical model of osteoporosis, Netrin-4, a product of vascular endothelial cells, could prevent bone loss by inhibiting osteoclast differentiation [158]. Thus, endothelial cells of specific tissues are differentiated in specific tissues and may perform essential functions in a paracrine relationship. However, care should be exercised in their study, as such endothelial cells are reported to de-differentiate in culture [159].

Thus, in such an integrated paracrine system, bioactive mediators in the blood first interact with endothelial cells in an organ or system and then evoke a response in those cells as well as passing through to interact with other target cells in those tissues (i.e., estrogen and progesterone first interact with endothelial cells and then osteoblasts and osteoclasts in bone). Part of the pattern of responsiveness regarding hormones such as estrogen are the estrogen receptors available in and on those cells (i.e., intracellular ER-alpha and –beta and cell surface variants) and perhaps their ratios.

Thirdly, the same tissues are not affected similarly in all females after puberty. Thus, the impact of puberty and subsequent variations in hormone levels during the menstrual cycle on the tissues of the female knee are heterogeneous. In studies of knee laxity variation during the menstrual cycle, ~80% of young females experienced changes in laxity during specific time periods in the menstrual cycle, but 20% did not, even though there were no detectable differences in hormone levels in the two populations [8,9,10]. As changes in laxity are the result of alterations in ligament function, the data indicate that ~20% of young females with regular menstrual cycles are resistant to such effects. Whether such resistance is associated specifically with the endothelial cells of the microvasculature or the ligament cells themselves remains to be determined. Interestingly, many females experience increases in ligament laxity during pregnancy. Whether this response pattern is due to sex hormones alone or in combination with other factors such as relaxin remains to be determined [160,161].

Thus, the unique features of the microvasculature and/or the innervation in tissues contributing to post-menopausal conditions and diseases could relate to the development of such conditions after menopause, rather than solely on other organ-specific target cells (i.e., osteoclasts and osteoblasts in bone) in the affected tissues.

## 6. What Potential Mechanisms Would Allow for the Development of Tissue-Specific Alterations in the Microvasculature and Innervation?

While it is possible that some of the risk of aspects of the conditions arising in the post-menopausal state have their basis in genetics and developmental programing during the time of sex determination and its consequences, it is also possible that the risk is imprinted at the time of puberty when levels for sex hormones become elevated and in the case of females, cyclically associated with menstrual cycles. Puberty may be an ideal time to imprint such risk via regulatory controls, as this would allow for post-natal growth and development and permit systems in the growing human to establish critical needs regarding regulatory controls in the adaptation to the environmental conditions of Earth, systems that likely preceded the evolution of humans. Thus, the bones and the heart and cardiovascular system need to adapt to the 1 g environment, the brain needs to establish its boundary conditions in the context of the geomagnetic field of Earth, energy regulation needs to balance growth and storage potential and the immune system needs to establish its various elements (i.e., the spleen, lymph nodes, mucosal locations) and relationships with the different microbiomes prior to further sex-specific regulatory controls.

As the cells of both males and females express receptors for both androgens and estrogens/progesterone [123] and both types of hormones, the mix of reagents and receptors is dramatically impacted on the tissues of both males and females at the time of puberty, and, thus, those changes that are related to some aspects of reproductive advantage could be imprinted at this time of change to many diverse systems, such are those related to reproduction but also the musculoskeletal system, cardiovascular systems and metabolic systems, to name a few. It should be noted that growth and maturation can lead to successful organ development, with the integration of functions between the cells involved, including any microvasculature and innervation regulatory functions, and, thus, puberty might impact successful templates for the tissues subsequently affected in the post-menopausal state.

If puberty is the time of imprinting, leading to subsequent post-menopausal risk, and the tissue-specific microvasculature/innervation systems are the targets, how could this imprinting be manifested? One strong possibility is via the epigenetic modification of the template in the affected tissues [162,163]. Not only are puberty-associated epigenetic modifications necessary for puberty to be successful [162,163], but a variety of tissues can be affected, likely in a tissue-specific manner. However, much research on this has been focused on the neuroendocrine regulation of reproductive aspects of puberty onset ([162,163] and others).

Thus, puberty is a time of dramatic epigenetic change in females and males, and a likely potential point of origin for the post-menopausal risk of disease due to a failure to maintain regulation with the loss of the production of the sex hormones that led to the epigenetic changes at puberty. Thus, puberty is potentially **WHEN** the potential for post-menopausal risk becomes evident.

The next important question to be asked is **HOW** such risk becomes manifest at the time of puberty in females via epigenetic modifications? In addition, there are some assumptions associated with the answers. First, one may have to assume that the epigenetic changes occurring at the time of puberty are not reversible after 35+ years of menstrual cycles. Secondly, additional epigenetic changes associated with life experiences between puberty and menopause do not impact the fidelity of the puberty-associated changes, although they could reinforce some of them, but that is a speculation.

There are a number of possible answers to the question of HOW the risk becomes manifested, some based on what is known regarding the epigenetic changes occurring in other contexts and some which are possibly more speculative. First, it is known that the promoters for a number of genes can become methylated at puberty [164,165]. Thus, changing the potential targets for ER/E and PR/P in a cell could lead to alterations in further gene regulation. Previous studies have indicated that ER in the absence of E can exert an influence on the expression of matrix metalloproteinase (MMP) genes in model systems [166,167,168,169], with differences noted between ER-alpha and ER-beta and also at the level of promoter variants for MMP, such as MMP-1. Thus, altering the expression of ER-beta or ER-alpha in individuals that were homogeneous for the 2G/2G variant of MMP-1 versus the 1G/1G variant could have an impact on MMP-1 expression and subsequent proteolytic events. Further demonstration that the activities of naked ER can influence gene expression comes from studies using deletion mutants of ER where the hormone binding domain for estrogen had been removed [166,170]. Thus, ER-alpha and ER-beta can be active mediators of gene expression in the absence of hormones. It is likely that receptors for other sex-related hormones function similarly.

While the above example used specific MMP, there are also other classes of proteinases that could be involved (i.e., serine proteinases or cathepsins) or other molecules. However, proteinases and their local dysregulation following menopause is an attractive focus, as they could mediate the loss of tissue or organ integrity at the level of endothelial cell-target cells via compromising the integrity of the unique extracellular matrix which maintains the integrity of each organ/tissue system. In addition, it is also likely that the regulation of genes other than proteinases may be epigenetically modified. For these genes and their modification pattern to be maintained throughout evolution, there must be at least some reproductive advantage, even if subtle and arising perhaps during the organ-specific differentiation of the endothelial cells in the organs affected by the post-menopausal dysfunctions. An additional potential implication of such a scenario is that during the time between puberty and menopause, the consequence of the epigenetic modifications is that the impacted tissues may become more sex hormone-dependent to retain integrity, and thus the loss of sex hormones at menopause could have an even more dramatic effect on system integrity.

The Potential Role of Epigenetics in Risk for Post-Menopausal Diseases/Conditions

Epigenetic modifications of the functioning of the inherited genome can take various forms, including the methylation of bases in the DNA [171] and alterations to molecules that bind to DNA (e.g., histones) [172]. These can occur at a variety of life stages from early life [173,174] and puberty [4] and via life experiences and exposure to environmental agents [175]. Many such epigenetic modifications are viewed as irreversible, but recently, an astronaut who was in space for ~1 year exhibited epigenetic changes in his peripheral blood white cells that were reversible after returning to Earth [176]. Thus, an individual inherits a genome with associated variations in the coding regions of genes, promoter elements and other regulatory elements, but the functioning of this inherited genetic material can be further influenced by epigenetic modifications.

While some risks of post-menopausal diseases or conditions may reside in the genome, an alternate hypothesis is that the majority of the risk arises during life transitions and may not manifest until after menopause. For most of evolutionary history, when the life expectancy for females was ~30–40 years of age, any risks arising early in life would likely not lead to consequences for any females living past 50. Based on this framing, and the above discussion, possibly the most likely life transition leading to post-menopause risk is puberty. That is the time in a female’s life when they undergo a transition to become ready for reproduction, with the onset of ovulation, menstrual cycles and adaptations for successful pregnancies and lactation.

The onset of puberty in females is accompanied by epigenetic alterations in many tissues, as discussed above. However, aspects of puberty can also be influenced by events occurring prior to puberty, such as epigenetic events resulting from exposure to environmental chemicals. While epigenetic changes accompanying puberty are well documented in some human cells or locations [162,163], the breadth and specificity of such changes in many different tissues has not been elucidated in humans. Thus, the focus has been more on the changes directed associated with the onset of reproductive integrity, often in preclinical models, than in other tissues perhaps more peripherally associated with the consequence of puberty.

By the time of puberty onset, an individual human has grown and matured at multiple levels. Thus, there has been intellectual and cognition growth as well as physiological growth. Organ systems have become integrated with regard to their multiple cell types, including microvascular cells and neural regulatory elements. Most organs and tissues are both innervated and vascularized, with the innervation often in close proximity to the vascular elements. Thus, the innervation and microvascular elements can form a regulatory complex in the context of each individual organ system. In female rabbits, it has been shown that in pregnant animals, this regulatory complex is altered during pregnancy [146,147].

There are at least three important aspects to the above information regarding this regulatory complex and its relationship to other cells in each tissue or organ. The first relates to the fact that all endocrine hormones and other molecules that are carried by the blood must first interact with endothelial cells of the microvascular before traversing to other target cells in an organ or tissue. While it is possible that some molecules could pass around the endothelial cells, many may also have to go through the endothelial cells. With tight junctions between endothelial cells, posing a “blood–brain” barrier in the brain but also a barrier in other organs and tissues, many molecules such as estrogen, progesterone and glucocorticoids have to traverse through the endothelial cells to get to other potential target cells. While traversing through the endothelial cells, which contain receptors for sex hormones [177,178], sex hormones could potentially influence the metabolism of the endothelial cells. Thus, at the time of puberty, the increase in sex hormones could lead to epigenetic alterations in the endothelial cells comprising the microvasculature of different organ systems.

Secondly, cells of the microvasculature are not all the same, and can vary in different locations such as the brain and individual organ systems [148,179,180]. Thus, during development, growth and maturation, and prior to the onset of senescence, the endothelial cells of the microvasculature of every organ and tissue likely becomes part of the integrated functioning of that identifiable component [148]. As such, it becomes adapted via some mechanisms (likely involving differentiation and possibly epigenetic modifications) to work in concert with the nerves and other cells comprising that tissue or organ. As such, during puberty, the onset of sex hormone surges could potentially impact the cells of different tissues differently, including at the epigenetic modification level. This concept opens up the possibility of incomplete or “faulty” epigenetic alterations to some tissues at the time of puberty, possibly with some genomic overtones, leading to their altered regulation after menopause as long as such alternations did not interfere with functioning during the time between puberty and menopause.

Alternatively, the epigenetic modifications occurring at puberty may be appropriate, but the variations influencing the post-menopause environment arise during development and are influenced by genetic variables at a time when the initial organ differentiation and subsequent maturation occurs. Either way, as post-menopausal diseases each affect only a subset of the female population and they may be polygenic, the error rate appears to be quite substantial for this population, if indeed the basis resides in error accumulation. This conclusion does have a caveat in that there is a focus on outcomes that are called diseases and there may be some variation in the mechanisms leading to the diseases. However, a central focus of all or nearly all of post-menopausal conditions may reside in the vascular component of the tissues or organs affected. Further study of this possibility may reveal new potential to address the underlying tenets of the diseases and subsequently new interventions.

Of note, while the focus of the present discussion is on menopause and the mechanisms for the onset of conditions/diseases after menopause, the mechanistic implications of the discussion may also pertain to an analogous process in males as they go through puberty and then andropause later in life. A lack of fidelity on initiating epigenetic changes at the time puberty could also lead to some diseases of aging in males.

Alongside the molecular basis for the risk and manifestation of post-menopausal disease/conditions is the finding, as discussed in the above sections, that in the case of many of these conditions they can be diminished in impact or somewhat prevented by initiating exercise protocols in midlife or during the progression of menopause. In addition to the effectiveness of exercise, it should be mentioned that menopause is a process leading to the loss of ovarian function and corresponding declines in sex hormone levels, and the onset of the postmenopausal conditions/diseases is also a process that can remain undetected until symptomatic. As such, the process is a form of “deconditioning” [65] that can be, at least partially, impacted or overridden by exercise. Thus, the commonality of the effectiveness of exercise on postmenopausal conditions could lead to an improved understanding of how the loss of hormones leads to vascular deconditioning and disease initiation in an organ/tissue-specific manner. Evaluation of the uniqueness of organ/tissue-specific endothelial cells from individuals of differing ages could provide insights into potential targets to address their involvement in post-menopausal diseases/conditions. However, one would have to make sure to avoid the dedifferentiation of the cells during in vitro culture [148,181].

## 7. The Challenge of Understanding Conditions Associated with Menopause from Those of Aging

While the above discussion has been focused on menopause and thus females, biological systems of both males and females change with aging [182,183,184]. Aging exhibits considerable heterogeneity in humans, with both chronological aging only part of the equation, and epigenetic clocks are also a consideration [185,186,187]. While all females appear to undergo menopause, a subset (~20%) of males undergo andropause [188,189]. Andropause is the gradual decline in testosterone levels in a subset of males, leading to the onset of a set of often mild symptoms, some of which can also be influenced by environmental factors [189].

Depending on the individual, aging can affect the integrity of a number of biological tissues and systems including the skin [190,191] and muscles [192,193], leading to sarcopenia in the latter tissue. Some aspects of the muscle changes can exhibit sex-specific differences during aging [194], and some of the changes in tissues such as the skin can also be affected by exposure to external factors such as sunlight and UV radiation [195].

Other tissues affected by aging are bone [196,197], with sex-differences being evident [197], and the cardiovascular system [198]. Interestingly, myokines such as irisin are released from active muscles and can also influence bone [199]. Thus, aging effects on muscles and their disuse could influence bone integrity as well, and, therefore, there could be overlap between the aging effects on these tissues. This issue could be addressed by interventions to prevent or diminish muscle loss with aging [200]. Sex differences in myocardial and vascular aging have also been reported [198], and, thus, there is an aging component to both cardiovascular and bone integrity as well as the menopause-associated changes in risk discussed previously for these tissues.

It is clear from the above discussion that the menopause-associated conditions or diseases discussed above occur in parallel with age-related alterations, with the latter occurring in an individual time frame. In addition, age-related alterations to several biological systems is complicated by the findings that aging is not just chronological but also appears to be influenced by aging parameters associated with epigenetic clocks, telomere shortening and other variables [187]. However, this interplay between menopause-related consequences and aging parameters could likely complicate an understanding of the molecular basis for the conditions arising after menopause.

## 8. Conclusions

Sex differences exist in a wide variety of biological and physiological systems, likely starting at the time of conception and progressing during development, and influence post-natal growth and maturation and the transitions during the onset of puberty. Many of these are the result of genomic influences as well as epigenetic modifications at key transition points, likely in preparation for successful reproduction. However, after menopause, some of these early influences appear to result in the appearance of diseases or conditions in different subsets of females (e.g., osteoporosis, obesity, dementia, cardiovascular disease and osteoarthritis). Much of the focus of research to date has been on the target cells in the various affected tissues (i.e., osteoblasts and osteoclasts in bone during osteoporosis development and progression). However, with such a focus on target cells, this may lead to the conclusion that the conditions are all separate entities. An alternative concept, however, is that, in addition to the target cells, there may be a commonality to the conditions that relates to the regulatory role of the endothelial cell system of the microvasculature that is uniquely differentiated in an organotypic manner [148]. The loss of the integrity of the paracrine regulatory function of endothelial cells after menopause could also serve as a disease-inducing mechanism. As the sex hormones that are lost during menopause (a process, not an event) likely traverse the endothelial cells of the affected tissues, the microvasculature of the different tissues affected by menopause may be a commonality in terms of the risk of the appearance of post-menopausal conditions due to early life changes that become manifest after menopause due to a loss of their regulatory input as an unintended consequence of much of evolutionary history. Thus, many post-menopausal conditions/diseases and disease risk may be the result of the loss of the regulatory influence of the tissue-specific endothelial cells of the microvasculature. If this hypothesis is proven to have merit, then it may open new directions for the development of improved interventions to prevent or diminish the impact of these post-menopausal conditions as well as provide a new understanding of how tissues are regulated. However, as post-menopausal changes are intercalated with those associated with aging and aging exhibits heterogeneity between individuals, any mechanistic overlap between those two processes will also need to be understood. Similarly, andropause will also need to be considered in the study of aging in males.

## Data Availability

This review did not involve any original data.

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
