# Peer review of "Sex Differences in Biological Systems and the Conundrum of Menopause: Potential Commonalities in Post-Menopausal Disease Mechanisms"

_ijms, 2022, doi:10.3390/ijms23084119_

Round 1
Reviewer 1 Report
This article on Sex Differences in Biological Systems and the Menopausal Conundrum: Potential Commonalities in Postmenopausal Disease Mechanisms presents solid information on the subject matter, but below is a series of recommendations that may provide higher quality at your work.
In general, the number of references is adequate for the length of the work, but they must be structured correctly, since there are paragraphs that have an excessive number of references and, on the contrary, other information does not have a bibliographic reference. Below are some examples. Please review the ENTIRE manuscript:
- Lines 31-37: the use of 8 references is excessive, so it is recommended to lower their number.
-Lines 37-41: this should be referenced. Please add relevant bibliography.
- Line 59: "[discussed in 13-15]" appears on this line. This bibliographic citation format is not correct. Please remove "discussed in". If you want to provide any relevant information on these works, you must include it in the text and refer only to the corresponding number. Review and correct the entire manuscript.
- Line 65: as in the previous case, remove "reviewed in" and add the relevant information in the text. Review and proofread throughout the manuscript.
- Lines 84-89: must be referenced.
- Lines 91-96: this information must be referenced.
- Lines 108-118: this information must be referenced.
- Lines 151-163: there is no reference to the information provided. Please add.
Lines 331-349: a paragraph with a great length and without any bibliographic citation. Please add.
The references must follow the format of the journal and have homogeneity, please review this section.
Reviewer 2 Report
I received a manuscript for review entitled: Sex Differences in Biological Systems and the Conundrum of Menopause: Potential commonalities in post-menopausal disease mechanisms. The manuscript is very interesting and brings a new perspective to the topic being described.
As a reviewer, I have a few comments:
There are many generalizations of sex in the manuscript. On the individual level the population is very heterogeneous.
lines 67 - 89. The author describes biological sex differences and function. There are few described differences between the female mare during this period. It would be useful to characterize other disease processes and individual features.
Lines 90- 105 Men are also at risk for hormone changes associated with aging. I propose to describe the differences between male and male aging.
The author describes: Osteoporosis, Dementia, Cardiovascular Disease, Obesity, Osteoarthritis as factors related to the aging of women. These subsections are fully described and correct. In my opinion, there are no chapters on atrophy of muscles, lung volume and skin aging.
Round 2
Reviewer 1 Report
All comments or suggestions have been made correctly, so I consider accepting the manuscript for publication.